# Modeling Alzheimer’s Disease: A Review of Gene-Modified and Induced Animal Models, Complex Cell Culture Models, and Computational Modeling

**DOI:** 10.3390/brainsci15050486

**Published:** 2025-05-05

**Authors:** Anna M. Timofeeva, Kseniya S. Aulova, Georgy A. Nevinsky

**Affiliations:** SB RAS Institute of Chemical Biology and Fundamental Medicine, Novosibirsk 630090, Russia

**Keywords:** Alzheimer’s disease, amyloid, presenilins, tau, animal models, transgenic mice, induced pluripotent stem cells, iPSC, spheroids, 3D scaffolds, microfluidics, computational modeling

## Abstract

Alzheimer’s disease, a complex neurodegenerative disease, is characterized by the pathological aggregation of insoluble amyloid β and hyperphosphorylated tau. Multiple models of this disease have been employed to investigate the etiology, pathogenesis, and multifactorial aspects of Alzheimer’s disease and facilitate therapeutic development. Mammals, especially mice, are the most common models for studying the pathogenesis of this disease in vivo. To date, the scientific literature has documented more than 280 mouse models exhibiting diverse aspects of Alzheimer’s disease pathogenesis. Other mammalian species, including rats, pigs, and primates, have also been utilized as models. Selected aspects of Alzheimer’s disease have also been modeled in simpler model organisms, such as *Drosophila melanogaster*, *Caenorhabditis elegans*, and *Danio rerio*. It is possible to model Alzheimer’s disease not only by creating genetically modified animal lines but also by inducing symptoms of this neurodegenerative disease. This review discusses the main methods of creating induced models, with a particular focus on modeling Alzheimer’s disease on cell cultures. Induced pluripotent stem cell (iPSC) technology has facilitated novel investigations into the mechanistic underpinnings of diverse diseases, including Alzheimer’s. Progress in culturing brain tissue allows for more personalized studies on how drugs affect the brain. Recent years have witnessed substantial advancements in intricate cellular system development, including spheroids, three-dimensional scaffolds, and microfluidic cultures. Microfluidic technologies have emerged as cutting-edge tools for studying intercellular interactions, the tissue microenvironment, and the role of the blood–brain barrier (BBB). Modern biology is experiencing a significant paradigm shift towards utilizing big data and omics technologies. Computational modeling represents a powerful methodology for researching a wide array of human diseases, including Alzheimer’s. Bioinformatic methodologies facilitate the analysis of extensive datasets generated via high-throughput experimentation. It is imperative to underscore the significance of integrating diverse modeling techniques in elucidating pathogenic mechanisms in their entirety.

## 1. Introduction

Alzheimer’s disease (AD) is a complex neurodegenerative disease characterized by progressive cognitive decline. This disease is defined by the pathological accumulation of insoluble amyloid β (Aβ) and hyperphosphorylated tau protein [1,2]. Extracellular Aβ aggregation is observed in the brain parenchyma and on vessel walls. The accumulation of hyperphosphorylated tau protein within neurons leads to the formation of neurofibrillary tangles (NFTs). Aβ plaques and NFT tubules are diagnostic hallmarks of Alzheimer’s disease, their aggregation resulting in neuronal dysfunction and mortality. Further investigation is required to fully elucidate the complex interaction between plaques and tubules [3,4].

The etiology of Alzheimer’s disease remains undetermined. A multitude of factors contributing to human vulnerability to this disease have been identified, including genetic and epigenetic influences, environmental exposures, and other elements [5]. The multifaceted etiology of Alzheimer’s disease presents significant challenges to early diagnosis and the development of disease-modifying therapies.

A classification of Alzheimer’s disease distinguishes between two forms: hereditary, also known as familial Alzheimer’s disease (FAD), and sporadic, also known as late-onset Alzheimer’s disease (LOAD). Less than 1% of Alzheimer’s cases are attributed to familial Alzheimer’s disease, which tends to manifest early. Autosomal mutations are present in multiple generations of affected families, with multiple members diagnosed with the disease. The genetic mutations linked to FAD are often found in genes that produce amyloid precursor protein (*APP*), presenilin 1 (*PSEN1*), and presenilin 2 (*PSEN2*) [6,7,8].

The genomic profile of the most common form of Alzheimer’s, sporadic Alzheimer’s, displays greater complexity. The pathogenesis of this form is not linked to a single gene but is shaped by a complex interplay of factors subtly affecting disease risk. The most significant of these is the apolipoprotein E (APOE) ε4 allele [9,10]. The presence of a single ε4 allele is estimated to increase the risk of Alzheimer’s disease by a factor of 2 to 4, while two ε4 alleles increase the risk by a factor of 8 to 16 [11,12]. A genome-wide association study (GWAS) has identified over 40 alleles from other genes that are associated with an increased risk of developing sporadic cases [13]. The roles of epigenetic, environmental, and lifestyle factors should also be considered.

No single model in the current literature successfully elucidates the pathogenic mechanisms of tau protein and Aβ in Alzheimer’s disease progression. A postmortem study of the human brain revealed that tau pathology precedes amyloid beta deposition with the increase in age [14]. This theory is supported by two pieces of evidence. First, dozens of trials with therapeutic agents targeting amyloid have failed to produce any clinical benefit despite marked reductions in amyloid pathology [15,16]. Second, a considerable proportion of older adults exhibit substantial Aβ plaque deposition without manifesting cognitive impairment [17]. Studies have demonstrated a significant correlation between the severity of cognitive impairment in Alzheimer’s disease and the extent of insoluble tau deposition, independently of amyloid burden [1].

Jack et al. [18,19] suggested another model for developing Alzheimer’s disease pathology. The pathogenesis of this model is characterized by the aberrant deposition of Aβ fibrils. Subsequently, neuronal degeneration commences, indicated by increased cerebrospinal fluid tau protein. Subsequent synaptic dysfunction is observed. The validity of this model has been established in multiple studies [20,21,22].

Multiple models of this disease are employed to investigate the pathogenesis, multifactorial etiology, and potential treatments of Alzheimer’s disease. An effective model should reproduce both the pathological features and symptoms of human disease and mimic the chronological sequence of pathological changes observed in vivo. A lack of suitable models for the early stages of Alzheimer’s disease is accounted for by the disease’s complex pathogenesis. The majority of extant models showcase the swift emergence of pathological occurrences and associated cognitive deficits mirroring the intermediate and advanced phases of the human disease trajectory.

Choosing a suitable model requires first defining the research problem. For example, existing mouse models reproduce Aβ plaques, neurofibrillary tubercles, or neurodegeneration, but different lines of mice tend to be used for this purpose. The impact of tau protein and Aβ aggregate distribution on neuronal function is examined through models utilizing the direct cerebral injection of these proteins. Studies on intercellular interactions in disease, focusing on cells with specific mutations, employ both two-dimensional and three-dimensional cell culture systems. While no current model constitutes a precise replica of human Alzheimer’s disease, their utility as research instruments is undeniable, provided appropriate selection based on the experimental objectives.

In this review, we consider various models of Alzheimer’s disease (Figure 1), including gene-modified and induced models of various animals, iPSC modeling, and computational modeling techniques. This is a narrative review that summarizes the main points and important information from sources with relevant information available in the PubMed database up to and including 2025. Existing reviews usually focus on only one of the methods for modeling Alzheimer’s disease. However, an in-depth study of the mechanisms of pathogenesis can only be achieved by combining different modeling methods. Integrating different modeling techniques facilitates the in-depth analysis and understanding of pathogenic mechanisms. This review aims to systematize the knowledge on modeling methods. It also considers modern approaches, such as the use of induced pluripotent stem cells (iPSCs), the creation of complex cellular systems, and the development of mathematical models. In contrast to animal models, these approaches are more human-centered, have great potential, and can replace some animal experiments.

The review is organized into the following sections.

Section 2 provides a brief overview of the genetic factors associated with the development of Alzheimer’s disease.

Section 3 describes transgenic animal models, including the most commonly used genetically modified lines of mice, rats, and other large, lower vertebrate, and invertebrate animals (zebrafish models, Drosophila melanogaster models, and Caenorhabditis elegans models).

Section 4 describes the induction of Alzheimer’s disease symptoms by the introduction of chemicals, proteins, and gene expression vectors.

Section 5 is devoted to human tissue-derived cell models. It describes in detail the iPSC technology, as well as approaches to the three-dimensional modeling of cell cultures, including a modern and promising method such as microfluidic systems.

Section 6 is dedicated to in silico methods. It includes the computational modeling of protein interactions, methods for analyzing gene and gene expression data, and omics technologies, as well as the use of artificial intelligence.

## 2. Genetic Factors Associated with the Development of Alzheimer’s Disease

The formation of amyloid plaques results from the deposition of insoluble Aβ. Amyloid plaque formation is hypothesized to commence with dysfunctional APP processing. APP is a transmembrane protein that undergoes extracellular cleavage by one of two activities, α- or β-secretase, resulting in the formation of large N-terminal extracellular fragments of secreted APP and smaller, membrane-bound C-terminal fragments. Subsequently, the membrane-retained C-terminal fragments are subjected to an intramembranous scission by the γ-secretase complex [23] to generate the APP intracellular domain (AICD) and simultaneously release p3 (after α-secretase cleavage) or amyloid β peptide (Aβ, after β-secretase cleavage).

The Aβ40 peptide was isolated from the cerebral vessels of Alzheimer’s patients and described in 1984 [24]. In 1992, it was shown that Aβ42 is the dominant form in the brains of patients with the familial form of the disease [25]. The characteristic pathology of Alzheimer’s disease involves the aggregation of less soluble Aβ42 peptides into fibrils, forming senile plaques (Figure 2A). It has been shown that the AICD is increased in the brain of people with Alzheimer’s disease and in animal models [26,27,28,29].

At least 40 APP mutations are documented as being linked to Alzheimer’s disease, with their impact on APP processing and Alzheimer’s disease development showing variation [30]. Nomenclature for these mutations is based on the geographic origin of the initially identified carrier family and the mutated residue within the longest isoform (APP_770_) (Figure 2B). The Swedish APP mutation (KM670/671NL) impairs the efficiency of β-secretase in cleaving the amyloid precursor protein, thereby increasing Aβ production and resulting in a 2–3-fold rise in plasma Aβ concentrations [31,32]. A number of mutations are associated with increased Aβ42 production and decreased Aβ40 formation through effects on γ-secretase activity. These mutations include Flemish (A692G) [33,34,35], Italian (E693K) [36], Dutch (E693Q) [37], Arctic (E693G) [38], Iowa (D694N) [39], Iranian (T714A) [40], Australian (T714I) [41,42], French (V715M) [42,43], German (V715I) [44], Florida (I716V) [45], London (V717I) [46], and Indiana (V717L) [47]. The heightened amyloidogenicity and aggregation propensity of Aβ42 compared with Aβ40 render individuals with these APP mutations more susceptible to Alzheimer’s disease. The Arctic mutation, E693G, does not affect either the total amount of Aβ or the ratio of Aβ42 to Aβ40. This mutation, however, results in an increased rate of aggregation for the mutant peptide [48].

Presenilins 1 and 2 (PS1 and PS2), integral components of the γ-secretase complex, play a crucial role in APP processing. Genetic mutations affecting these proteins result in increased amyloidogenic Aβ42 peptide production, a contributing factor to familial Alzheimer’s disease [49]. To date, more than 200 unique PSEN1 gene mutations and 16 PSEN2 gene mutations linked to Alzheimer’s disease have been documented [50].

A primary genetic risk factor for sporadic, late-onset Alzheimer’s disease is the ε4-type allele of the APOE gene [51]. Additionally, mutations have also been identified in other genes associated with Alzheimer’s disease, such as the NOTCH3 gene [52], SORL1 [53], SNP [54], TREM2 [55,56,57], and others. Extensive research has investigated genetic variants associated with Alzheimer’s disease [58,59,60,61,62,63,64]. It should be noted that the identified variants are predominantly characterized by low frequency [65,66,67].

The overwhelming majority (exceeding 99%) of Alzheimer’s disease cases are of the sporadic type and thus not associated with any recognized genetic mutation. Given the idiopathic nature of Alzheimer’s disease, it is likely that multiple factors contribute to its pathogenesis. These may act concurrently or sequentially to initiate the disease cascade, with one factor initiating the disease and others driving its progression. The multifaceted character of this disease presents a considerable challenge to modeling efforts.

## 3. Transgenic Animal Models

The protracted clinical course of Alzheimer’s disease, characterized by extensive neuropathological changes, presents considerable difficulties in the development of accurate models. Numerous experimental animal models that reproduce some features of Alzheimer’s disease have been described in the literature. Model species diversity is extensive, encompassing invertebrates to primates, with a predominance of murine models.

### 3.1. Mouse Models

The utilization of mice in research benefits from their brief reproductive cycles, economical maintenance, and well-defined experimental procedures [68]. A wide array of tools and protocols exist for murine experimentation, encompassing standardized assessments of learning, memory, behavior, and cognition [69,70,71,72].

Over 200 distinct genetically modified murine lines exhibiting Alzheimer’s disease-related phenotypes have been documented. For example, the resource alzforum.org [73] (accessed 23 October 2024) provides information on the characterization of 228 models of Alzheimer’s disease (Figure 3). This resource organizes information on the characteristics of different animal models of Alzheimer’s disease and allows for the selection of appropriate mouse and rat strains for study.

The models comprise transgenic mice genetically modified to express human genes with one or more Alzheimer’s-related mutations (APP, PS1, PS2, MAPT, and ApoE4) [74,75,76]. Mouse lines knockout for these genes have also been created [50]. This review considers a limited number of the developed mouse lines. For more detailed information on the current models, one is referred to several reviews [50,68,77,78,79,80].

The initial transgenic mouse model exhibiting amyloid pathology was created by D. Games [81]. Transgenic mice overexpressing human amyloid precursor protein (APP) harboring the Indiana mutation (AβPP_Ind_) display age-dependent increases in amyloid beta plaques alongside cognitive and learning impairments [82].

The Swedish mutation serves as a common tool for producing models characterized by Aβ overexpression. Notably, the Tg2576 line (AβPP _Swe_ × PS1 _M146L_ [83]) exhibits a 5-fold augmentation in Aβ40 and a 14-fold augmentation in Aβ42 production, in addition to the presence of numerous plaques and a range of behavioral, biochemical, and pathological features analogous to those observed in Alzheimer’s disease [84].

Additionally, mouse lines carrying two mutations in the APP gene have been created. These are TASD [85] mice, with the Swedish and London mutations, and J20 [86] and TgCRND8 [87] mice, with the Swedish and Indian mutations. Mouse models with double mutations, especially TgCRND8 mice, exhibit an earlier onset of pathology compared with mouse models with a single mutation in APP.

The development of biogenic mouse models for Alzheimer’s disease has involved both the crossing of distinct transgenic lines and the addition of further transgenes to pre-existing lines. This methodology yielded the transgenic lines AβPP _Swe_ × PS1 _M146L_ [83], AβPP _Swe_ × PS1 _A246E_, AβPP _Lnd_ × PS1 _A246E_, and AβPP _Swe_ × PS1 _ΔE9_ [88]. The Alzheimer’s disease pathology in bigenic mice is characterized by a markedly earlier onset and more rapid progression than that observed in monogenic mice.

A triple-transgenic mouse model (3xTg) incorporates human mutations of AβPP _Swe_ × PS1 _M146V_ × tau _P301L_ [75]. Notably, 3xTg mice demonstrate the formation of both amyloid plaques and NFTs within their cerebral structures. Among the most frequently utilized models for studying Alzheimer’s disease, the 3xTg model is prominent in research on tauopathy and amyloid pathology development [89,90,91].

Among murine models of the disease, the 5xFAD line exhibits the most rapid disease progression. The model integrates APP mutations from Swedish, Floridian, and London sources, along with the PSEN1 gene mutations M146V and L286V. Efficient transgene expression is achieved under the control of the murine Thy-1 promoter [76]. Consistent with expectations, the 5xFAD model exhibits severe pathological features. At 1.5 months, extracellular amyloid beta accumulation and senile plaques are evident, and neuronal populations are depleted by 2 months of age. Furthermore, these mice exhibit cognitive deficits at an earlier stage compared with other murine models [92,93].

Mice with ApoE gene mutations associated with sporadic disease presentation have been generated. Nevertheless, the APOE ε4 allele is insufficient to induce a complete Alzheimer’s-like disease phenotype, manifesting only in isolated pathologies [94,95]. The literature describes transgenic models exhibiting мyтaциями AβPP _Swe_ and PS1 _ΔE9_ mutations and expressing APOE ε2 или APOE ε4 [96]. Thy1-ApoE4/C/EBPβ double-transgenic mice were developed, exhibiting strong ApoE4-mediated activation of the transcription factor C/EBPβ and subsequent augmentation in δ-secretase activity. This mechanism contributes to increased APP and tau cleavage and the development of Alzheimer’s disease-like pathologies [97]. Transgenic mouse studies have demonstrated that ApoE is essential to the development of fibrillar amyloid plaques [98].

In the mouse models previously discussed, Alzheimer’s disease-associated Aβ pathology is the primary feature, while tau pathology is not observed. The modeling of neurofibrillary pathology in murine models typically involves the expression of transgenic human tau, frequently combined with other transgenes [99,100]. However, it must be emphasized that these tau mutations are not causative in Alzheimer’s disease; thus, a discussion of mice with these mutations has been omitted from this review. For more detailed information, one is referred to other papers [75,99,100,101,102,103].

The overexpression of human transgenes, driven by a variety of promoters, is a common feature of transgenic models, yet this methodology does not recapitulate the physiological expression profiles observed in mice. Furthermore, the genetic sequences of mice and humans exhibit considerable divergence. Hence, the use of knockin mouse models has extended to the modeling of Alzheimer’s disease mechanisms. Specifically, lines of mice expressing a humanized *App* sequence, further modified to include mutations characteristic of familial Alzheimer’s disease (FAD), have been generated. For example, the APP^NL^, APP^NL−F^, and APP^NL-G-F^ lines were obtained this way [104,105]. Furthermore, the creation of bi- and tri-genic models employs knockin mouse lines featuring humanized or mutant genes such as *Mapt*, *Psen1*, and *Apoe* [75,106,107]. In addition, investigations into the roles of genes linked to Alzheimer’s disease have employed gene knockout mouse lines, exemplified by *Apoe* and *Trem2* [108,109].

The development of mouse models for the sporadic, late-onset form of Alzheimer’s disease is currently in progress. No human gene mutations have been found to exert a comparable influence on the pathogenesis of this form of the disease as in familial Alzheimer’s disease. The MODEL-AD consortium has developed several models, including LOAD1 and LOAD2, to analyze the diverse genetic and environmental factors contributing to sporadic cases of the disease [110,111]. LOAD1-based knockin models have been created that include not only the original humanized APOEε4, the *Trem2*^R47H^ mutation, but also one of the mutations identified by a full-genome association search and inserted into the corresponding genes (e.g., *Abca7*, *Mthfr*, *Plcg2*, and *Sorl1*) [112].

Despite partially mimicking Alzheimer’s disease pathology, no existing model comprehensively reproduces all features of the human disease. Despite limitations, current animal models offer a potential avenue for investigating crucial aspects of disease pathophysiology that are inaccessible via human studies. A thorough understanding of the selected model and its constraints is crucial to the accurate interpretation of experimental results and extrapolation to human Alzheimer’s disease. Inter-species differences between rodents and humans influence the cleavage and biochemical properties of human Aβ in transgenic rodent models. The solubility of Aβ plaques in transgenic murine models surpasses that observed in human Alzheimer’s disease, a disparity likely attributable to the absence of post-translational modifications in murine Aβ, such as N-terminal degradation, isomerization, racemization, pyroglutamylation, and oxidation [113]. It is imperative to account for this factor, as illustrated by the markedly improved efficacy of Aβ clearance drugs in transgenic mice compared with human subjects, owing to differences in solubility.

### 3.2. Rat Models of Alzheimer’s Disease

In terms of physiology and genetics, rats demonstrate a higher degree of similarity to humans in comparison with mice [114,115]. The larger body and brain size of these subjects improves their utility in a variety of experimental paradigms, encompassing neuroimaging, intrathecal drug administration, microdialysis, repeated sampling, and in vivo electrophysiology [116]. Additionally, the more complex behaviors of rats allow for a greater quantity of behavioral measurements. Nonetheless, the progress in developing transgenic rat models has been hampered by their reduced reproductive rate and increased spatial requirements [117].

Common transgenic rat models, similar to mouse models, utilize *APP* and *PSEN1* transgenes containing mutations representative of inherited Alzheimer’s disease. The transgenic rat model Tg478/Tg1116 served as the initial model, characterized by the presence of Swedish and Swedish-London mutations in the *APP* gene. Amyloid plaques were observed in these rats at 17–18 months of age [118]. Amyloid plaque development was observed at nine months of age in a congruent rat model of PSAPP (Tg478/Tg1116/Tg11587), in which Tg478/Tg1116 rats also expressed PSEN1M146V [118,119]. However, these rats lack neurofibrillary pathology and neuronal loss. Furthermore, premature mortality in PSAPP rats is likely attributable to genetic abnormalities stemming from the introduction of three transgenes [119].

The McGill-R-Thy1-APP transgenic rat line, harboring the AβPP _Swe_ × AβPP _Ind_ mutations, provides a more comprehensive model of amyloid pathology, closely resembling Alzheimer’s disease. Amyloid beta plaques become detectable in the brains of these animals at six months of age. The developmental cognitive impairment characteristic of this model are demonstrably present from three months of age and have been extensively described [120].

Amyloid beta accumulation in the TgF344-AD bigenic rat model carrying AβPP _Swe_ × PS1 _ΔE9_ exhibits a strong correlation with age [121]. At the age of sixteen months, these rats display Gallyas-positive structures. These structures, similar in nature to NFTs, have not been previously characterized in other transgenic rat or mouse models. In addition, neurodegenerative changes in the cortex and hippocampus are observed at the age of sixteen months within this line.

Efforts to generate rat knockin lines expressing humanized and mutant *APP* and *PSEN1* failed to produce amyloid and tau pathology throughout the lifespan of the animals [122,123].

In summary, transgenic rats show promise as a research tool in Alzheimer’s disease. Notwithstanding certain advantages, the practical application of such models remains infrequent.

### 3.3. Large Animal Models

The search for improved human-applicable models has led to a focus on larger animal models, especially those exhibiting closer evolutionary proximity to humans and increased longevity [124]. As evidenced by [125], amyloid plaques and cerebral amyloid angiopathy were found in aged great apes [126,127,128]. It is important to note that these plaques show a largely diffuse pattern and occur with less frequency than in cases of human Alzheimer’s disease. A substantial distinction exists between great apes and humans and other model animals in their elevated tau sequence homology. Notably, sequence identity rates of 100% and 99.5% exist between the human tau protein and that of chimpanzees and gorillas, respectively [129]. The creation of NFTs by great apes, though possible, remains an exceptionally rare phenomenon [128]. Due to ethical considerations, the use of great apes as models for human disease is infrequent [130].

Old World monkeys, members of a separate primate parvorder, are the most frequently employed animal models. For example, these are rhesus macaques (*Macaca mulatta*), Javan macaques (*Macaca fascicularis*), baboons (*Papio cynocephalus*), vervet (*Chlorocebus pygerythrus*), and marmoset (*Callithrix jacchus*). The rhesus macaque is a more frequently utilized research subject. Primates exceeding 25 years of age demonstrate the presence of amyloid plaques with a human-characteristic distribution [131,132]. Cerebral cortical Aβ accumulation increases with age, reaching concentrations comparable to those observed in human Alzheimer’s disease [133]. From a husbandry perspective, the reduced size and lifespan of marmosets offer considerable practical benefits. Marmosets exhibit physiological and neurological similarities to humans, including comparable cognitive and social behaviors [134,135]. Human and marmoset Aβ share an identical amino acid sequence.

High sequence homology of tau is observed in humans, rhesus macaques, and marmosets, all exhibiting hyperphosphorylated tau. Nevertheless, neurofibrillary tangle formation is predominantly absent in marmosets and rhesus macaques [136,137]. Advanced age in other primates, specifically baboons, is associated with significant albeit regionally restricted hippocampal tauopathy affecting 90% of animals older than 26 years [138].

Primates are presently the primary subjects in investigations into the toxicological impact of Aβ, encompassing Aβ injection protocols. This constitutes an effective instrument for examining the role of Aβ in neuronal dysfunction and neurodegeneration [139].

A model of Alzheimer’s disease in green monkeys (*Chlorocebus sabaeus*) was generated through gene-editing technologies, specifically by deleting exon 9 of the *PSEN1* gene [140]. A transgenic Javan macaque model was generated through the introduction of *hAPP*, incorporating Swedish, Arctic, and Iberian mutations. This yielded a substantial elevation in the Aβ42/Aβ40 ratio [141]. Marmoset models, transgenic for PSEN1 _C410Y_ and PSEN1 _A426P_, demonstrated significantly increased plasma Aβ42 concentrations [142].

Pigs constitute another large animal model for Alzheimer’s disease research. Two transgenic porcine models of Alzheimer’s disease were generated by utilizing miniature swine. The initial model incorporated the AβPP _Swe_ transgene. In these animals, however, no evidence of memory dysfunction was found [79]. The second miniature pig model contained triplicate AβPP _Swe_ and single PSEN1 _M146L_ transgenes. The pigs in the study exhibited intraneuronal accumulation of Aβ42 [143].

### 3.4. Lower Vertebrate and Invertebrate Animal Models

Large-scale genetic and chemical screenings in large animals present significant challenges. Models of reduced complexity are more appropriate for this form of analysis. Simple model organisms present advantages in terms of cost efficiency, high-throughput screening capacity, ease of genetic modification, and access to mutant and transgenic lines. Therefore, research utilizes invertebrate animal models (*Drosophila melanogaster* and *Caenorhabditis elegans*), as well as lower vertebrate models (*Danio rerio*). *D. melanogaster* and *C. elegans* have short lifespans, making it possible to study aging. The simple neuroanatomy of *C. elegans* allows individual neurons to be studied in their biological context [144]. Behavioral assays performed on *C. elegans* and *D. melanogaster* facilitate the evaluation of neuronal function and the detection of dysfunction that anticipates the physical signs of neurodegeneration. The zebrafish (*Danio rerio*) constitutes a simplified vertebrate model system that surpasses the practicality of mice and rats in screening procedures. Nevertheless, the application of invertebrate and lower vertebrate animal models is constrained by their limited homology to humans. Orthologs of several genes implicated in Alzheimer’s disease are expressed in *D. melanogaster*, *C. elegans*, and *D. rerio*, although notable differences exist between these orthologs and their human homologs [144,145,146]. Regions of genes significantly involved in Alzheimer’s disease pathophysiology are often absent in invertebrate orthologs. Consequently, diverse genetic modifications facilitating human transgene expression are employed for disease modeling.

#### 3.4.1. Zebrafish Models

*Danio rerio*, commonly known as the zebrafish, is a freshwater fish species that currently ranks second in popularity as a biomedical research model, following mice and rats. Zebrafish species are regarded as a promising tool for studying neurodegenerative diseases. Key advantages of zebrafish include their high reproductive rate, rapid development, and low maintenance costs [147,148]. Moreover, the straightforward nervous system of *Danio rerio* allows their behaviors to be quantified. Several assessment tools exist for evaluating spatial memory deficits, associative memory impairments, and anxiety levels in subjects of this species [149]. A range of zebrafish strains, encompassing AB, Casper, Tübingen (TU), Tüpfel long fin (TL), and Ekkwill, are utilized within the research community [150]. The experiment should account for the distinct characteristics inherent in each strain. The sequencing of the *Danio rerio* genome has been completed [151]. There is approximately 70% homology between the genomes of *Danio rerio* and *Homo sapiens*. Notably, genetic analysis has revealed that these fish possess genes associated with human diseases such as Alzheimer’s disease [146].

Genes associated with dementia in humans are homologous to the zebrafish genes, including the co-orthologs PSEN1 (*zf-ps1* [152,153], PSEN2 (*pre2* [153,154], APP (*appa* and *appb*) [155], and *MAPT* (*mapta* and *maptb* [156]). Danio rerio gene knockdown, targeted mutagenesis, and transgenesis have been previously reported [157]. One common method for transient gene knockdown in zebrafish is morpholino antisense oligonucleotide injection [158]. A similar approach has been employed to study the functions of these gene orthologs [159,160].

A number of zebrafish models have been developed through the introduction of human genes linked to Alzheimer’s disease. For instance, the introduction of the APP _Swe_ gene resulted in morphological alterations in the brain, histopathological changes within the vasculature, and amyloid beta accumulation. Transgenic tau *Danio rerio* models for Alzheimer’s disease have successfully demonstrated neurofibrillary tangles, neuronal loss, and cell death similar to the expected human pathology [161,162,163].

In addition, chemically induced models of Alzheimer’s disease symptoms were created on Danio rerio by using okadaic acid, aluminum chloride, and Aβ42 [164,165,166,167].

For primary whole-animal toxicity studies, the zebrafish exhibits advantageous characteristics, namely, facile drug administration and considerable toxin sensitivity, making it a superior vertebrate model. The efficacy of systemic dermal administration and targeted delivery of pharmaceuticals in aquarium water has led to the widespread and successful use of *Danio rerio* in research. For example, *Danio rerio* models have been used to test the effects of drugs used to treat Alzheimer’s disease [165,168,169].

*Danio rerio* offers a valuable model for investigating age-related cognitive and neurobiological alterations. Targeted manipulations can induce age-related phenotypes in Danio rerio that mirror those of aging humans [147]. In addition, the predictive capacity of gene network reconstruction methods [170] for molecular genetic processes is discussed extensively in Section 6. Integrating transcriptomic and interactomic data, alongside a comparison with mammalian data, could enhance our understanding of Alzheimer’s disease pathogenesis [171].

It can be concluded that *Danio rerio* is a promising model and tool for elucidating the processes underlying aging and Alzheimer’s disease.

#### 3.4.2. *Drosophila melanogaster* Models

*D. melanogaster* has proven to be a valuable model organism for investigating gene interactions in neurological disorders such as Alzheimer’s disease. The assessment of impacts in transgenic flies is achieved through diverse screening methods, encompassing the histological analysis of brain structures, phenotypic eye analysis, behavioral testing (locomotor and associative learning), and longevity studies [172,173]. The effects of gene expression on pathology development in *D. melanogaster* can be evaluated through RNA interference mechanisms, possibly serving as a precursor to research on more closely related human model organisms [174].

The absence of Aβ peptide formation in typical flies is attributable to a lack of β-secretase activity and sequence discrepancies between Appl and its human ortholog APP [175]. Transgenic flies were created that carry constructs encoding both human APP and human β-site APP cleavage enzyme 1 (i.e., β-secretase). Specific observation revealed Aβ plaque deposition in the retina and age-related neurodegeneration. Pervasive gene expression was associated with a shortened lifespan and defective wing vein development. These APP-based models prove useful for screening genes, drugs, or metabolites [176].

Human tau-overexpressing transgenic models of *D. melanogaster* have been generated [177]. These models are employed to explore the molecular mechanisms of neurodegeneration within neurofibrillary pathology.

*D. melanogaster* models of Alzheimer’s disease offer advantages due to their comprehensively understood genome, short lifespan, and capacity for high-throughput analysis. However, it is important to take into account the differences between vertebrate and invertebrate organisms, including in the functioning of neurotransmitters [178].

#### 3.4.3. *Caenorhabditis elegans* Models

Alzheimer’s disease research has employed the well-characterized 302-neuron neural system of the soil nematode *Caenorhabditis elegans* as a model. Numerous *C. elegans* strains have been engineered to mimic the human pathology of Alzheimer’s disease [179,180]. Modeling Alzheimer’s disease in *C. elegans* has primarily focused on the transgenic expression of human Aβ and tau [181]. Nevertheless, the singular insertion and expression of human APP in *C. elegans* induces neurodegeneration and neurobehavioral deficits [182]. Models expressing human Aβ peptide in specific cell types were created. Age-dependent paralysis was observed in nematodes expressing the Aβ42 peptide in their body wall muscle cells [183].

Aβ peptide length is correlated with its degree of toxicity. Prior research indicated that Aβ42 exhibits higher toxicity compared with Aβ40. Research using *C. elegans* has demonstrated that Aβ38 can alleviate the phenotype resulting from Aβ42 expression in the GMC101 strain [184]. The findings indicate a potential neuroprotective effect of certain Aβ variants, a factor with significant implications for understanding the therapeutic limitations of indiscriminate Aβ peptide targeting.

Multiple fascinating models have been devised for *C. elegans* to examine oligomer formation and its implications. One of them is an optogenetic model based on the expression in *C. elegans* of Aβ with a fluorescent marker that rapidly oligomerizes in the presence of blue light [185]. An additional significant approach to studying oligomerization involves the generation of oligomer-specific antibodies, enabling the precise quantification of oligomers in vitro and within the *C. elegans* GMC101 model [186]. This model exhibited severe physiological defects, neuronal dysfunction, and neurodegeneration as a consequence of Aβ expression. By utilizing fluorescence lifetime imaging microscopy (FLIM), the quantification of amyloid fibril formation as a function of age has been accomplished [187].

In *C. elegans*, *ptl-1* encodes a tau-like protein that shares 50% homology with mammalian tau [188]. The loss of *ptl-1* shortens lifespan, impairs tactile sensitivity, and causes abnormal morphology in tactile neurons, replicating some aspects of Alzheimer’s disease pathology [189,190]. There are many *C. elegans* models overexpressing either wild-type or mutant forms of human tau protein, with the latter showing greater toxicity [191,192,193,194,195].

## 4. Induced Models

A range of agents are utilized to experimentally induce Alzheimer’s disease in animal models. The substances in question include numerous proteins and chemicals integral to the disease’s pathogenic process. The administration of chemicals, including but not limited to streptozotocin, scopolamine, colchicine, and okadaic acid, may result in neuroinflammation and neurodegeneration. The pathogenic peptides and proteins Aβ42 and tau are etiological factors in the symptomatology of Alzheimer’s disease. The administration of disease-inducing substances can be accomplished by direct delivery to particular brain regions (intrahippocampal and intracerebroventricular), as well as through oral and intraperitoneal methods [196]. The simultaneous disease onset in a large animal cohort represents a key advantage of such models, enabling the assessment of compound effects within isolated brain regions. Induced models offer a valuable addition to the current understanding of animal models and may contribute to the further elucidation of Alzheimer’s disease pathogenesis.

### 4.1. Induction of Alzheimer’s Disease Symptoms by Chemicals

Okadaic acid (OKA) is a prominent polyketide toxin known for its selective inhibitory action on serine/threonine phosphatases 1 and 2A [197]. Protein phosphatase 2A activity has been shown to be reduced in Alzheimer’s disease [198], leading to the hyperphosphorylation of tau [199]. Studies have shown that the intracerebroventricular infusion of OKA (70 ng/day) in rats results in the manifestation of Alzheimer’s disease-related pathologies, namely, tau hyperphosphorylation, apoptosis, and the cortical accumulation of nonfibrillar Aβ [200]. Subsequent research revealed the onset of Alzheimer’s-like cognitive impairment in rats 15 days following the intracerebral administration of 200 ng OKA [201]. Therefore, it is possible for the injection of OKA to be used to create an animal model of Alzheimer’s disease.

Likewise, administering a substantial colchicine dose can result in Alzheimer’s disease-associated pathologies, including cognitive deficits and behavioral alterations [202]. The impact of colchicine includes the disruption of axoplasmic transport and subsequent severe damage to hippocampal granular cells and mossy fibers, which culminates in neuronal loss, cognitive impairment, and spontaneous motor behavior in animal subjects [203]. Colchicine therapy may also induce microtubule disruption, a pathological feature analogous to that observed in Alzheimer’s disease [204]. It is tau dephosphorylation, and not hyperphosphorylation, that underlies the mechanism of action of colchicine [205]. The chronological relationship between pathological events and cognitive dysfunction has yet to be explored using this model.

Scopolamine, a muscarinic acetylcholine receptor (mAChR) antagonist, blocks acetylcholine transmission at the synapses upon administration. Animal studies show that scopolamine causes memory and learning deficits by damaging cholinergic neurons in the hippocampus [206]. Alzheimer’s disease is also characterized by substantial cholinergic neuron loss [207]. This process is modeled by using intraperitoneal and intracerebroventricular scopolamine administration. Administration of this substance to rats (1 or 2 mg/kg daily intraperitoneally) has been shown to increase the level of Aβ peptide and the level of APP expression in the animals’ brains, as well as increasing the level of phosphorylated tau [208,209].

Streptozotocin, a molecule formed from glucosamine and nitrosourea, causes pancreatic beta-cell death through its interaction with the glucose transporter GLUT2. Low intracerebroventricular doses of the substance damage the hippocampus and other structures crucial to learning and spatial memory, leading to impaired function. Javan macaques injected with streptozotocin (2 mg/kg) intracerebroventricularly showed signs of brain atrophy, astrogliosis, microglial activation, amyloid beta buildup, and tau pathology [210]. This model is considered to be a model of sporadic Alzheimer’s disease.

A number of other compounds can be employed to represent specific aspects of Alzheimer’s, such as cognitive deficits, oxidative stress, inflammation, and neuronal loss [196,211].

### 4.2. Stereotactic Administration of Various Forms of Tau and Aβ

Stereotactic drug delivery is a surgical procedure that uses a three-dimensional coordinate system to define the location and execution of actions in the brain. The exact location of the target area is described by using reference points on the skull and stereotactic coordinates [212]. The coordinates are usually determined by using rodent brain atlases [213] or online tools such as that in [214].

The stereotactic injection of different tau and amyloid types into specific brain regions enables the evaluation of their neurotoxic effects and their interplay with endogenous peptides and proteins in wild-type or transgenic models. As demonstrated in [215], the injection of synthetic Aβ42 fibrils into transgenic mice carrying a P301L tau protein mutation initiates tau pathology. These findings corroborate the amyloid hypothesis concerning Alzheimer’s disease etiology. The prion-like distribution of tau protein and Aβ aggregates across brain structures, as observed in injected models, demonstrates functional connectivity among the tau protein structures [216,217,218].

Synaptic dysfunction is a significant factor in the pathogenesis of Alzheimer’s disease [219]. Stereotaxic injection experiments utilizing diverse tau and Aβ variants have elucidated their cytotoxic mechanisms and roles in synaptic impairment [220,221,222]. Low-molecular-weight oligomeric prefibrillar Aβ aggregates have been shown to be the most cytotoxic forms [223], impair long-term memory after injection into mice [224,225,226], impair synaptic plasticity, and cause cognitive impairment in rats [227]. Concurrently, the bilateral intrahippocampal administration of fibrillar Aβ resulted in decreased neuronal density and concomitant behavioral impairments [228,229].

Despite its intracellular localization, various tau species are demonstrably present within cerebrospinal fluid and the media of cultured neuronal cells [230]. Studies indicate a paradigm shift in understanding tauopathies, focusing on soluble tau aggregates and their role in synaptic disruption rather than the previously emphasized insoluble fibrillar forms [231,232]. Wild-type tau and the P301S mutant tau, known for its increased aggregation propensity, when assembled into filaments and introduced into the brain ventricle, were defined to exhibit a marked lack of reactivity [233,234]. Soluble tau aggregates, in contrast to fibrillar tau, elicited the swift suppression of long-term hippocampal potentiation following injection. The exogenous introduction of soluble tau aggregates and oligomers, derived from the brains of Alzheimer’s disease patients, into animal models induced cognitive impairment in mice, a phenomenon potentially explained by in vitro evidence of impaired hippocampal synaptic plasticity [235,236]. The presented data, along with other findings, imply a correlation between specific diffuse tau conformations and synaptic dysfunction within the context of Alzheimer’s disease.

In comparison with transgenic models, neurotoxic peptide and aggregate administration leads to rapid symptom onset in animals (within weeks), which is advantageous for evaluating the efficacy of various compounds [157,237]. The accurate administration of tau protein and Aβ to animals demands a certain level of skill. Furthermore, the injection process can induce inflammation at the injection site, a factor that necessitates careful consideration during data interpretation.

### 4.3. Stereotactic Delivery of Vectors for Expression of Genes Associated with Alzheimer’s Disease

Recombinant viral vector technology has proven to be an essential tool within neuroscience for the manipulation of neural function both in vivo and in vitro. In neurobiology, adeno-associated viruses (AAVs), retroviruses, and lentiviruses represent the three most prevalent viral vectors employed for gene delivery [238]. Each viral type presents both strengths and weaknesses. The absence of chromosomal integration of AAV-delivered genes, resulting in their existence as concatemeric circles, ensures prolonged transgene expression [239]. For this reason, adeno-associated viruses are the most commonly employed viral vectors for gene therapy. A primary limitation of AAVs is their modest packaging capacity, restricted to a maximum of 5.2 kb of genetic material [240]. The genetic cargo capacity of lentiviruses is significantly higher, reaching a maximum of 9 kb. Upon viral entry, the lentiviral RNA genome is reverse-transcribed into DNA and integrated within the host chromosome at random locations. Insertional mutagenesis, a considerable drawback of retro/lentiviral vectors, can lead to compromised cellular function and/or cell death [241,242].

Viral vectors are becoming increasingly prevalent tools for inducing neuropathology, either independently or synergistically with established genetic models. Initially, models utilizing APP and Aβ sequence transduction were generated. Following the intrahippocampal administration of AAV1 vectors expressing Aβ42 in adult Wistar rats, amyloid deposition was detected after a four-month period [243]. Further research involving the prefrontal cortical injection of AAV vectors with diverse APP gene mutations in adult mice produced amyloid and neurofibrillary tangle pathology, accompanied by microgliosis and reactive astrogliosis [244]. The hippocampal administration of a lentiviral vector containing mutant human amyloid precursor protein induced amyloid beta deposition, astrogliosis, and memory impairment in rats [245].

Both lentiviral and AAV vectors have been used to express different forms of tau in the brain in mice and rats, and both can generate hyperphosphorylated tau in vivo. Pathological hyperphosphorylation was observed in both wild-type and mutant (P301L or P301S) tau following lentiviral vector transduction. The mutant tau, however, demonstrated a more severe phenotype [246]. NFTs were obtained when tau with the P301S mutation was transduced by using a lentiviral vector into the APP23 mouse line carrying APP with the Swedish mutation [247]. An investigation employing the neonatal intracerebral ventricular administration of AAV1-tau-P301L in wild-type mice revealed tau hyperphosphorylation, subsequently leading to NFT development [248].

Viral transgenesis offers advantages compared with traditional transgenic models. First, it is less costly and faster to establish than germline manipulation. Second, the pathology can be regionally targeted, allowing neural projections to be analyzed and peripheral expression to be avoided. Third, the same virus can often be used in mice or rats, young or old animals, and for any region of interest. Fourth, viral inoculation can be combined with genetic models to test functional interactions more quickly and easily than crossing multiple alleles. However, these models also have a number of drawbacks. It should be noted that transduction exhibits mosaicism: not all cells are transduced, and the number of viral particles internalized and the level of subsequent expression differ among transduced cells. Moreover, given that surgical injection and/or viral transduction trigger an injury response, the observed phenotypes could reflect a complex interaction between the expressed viral transgene and this cellular response. Therefore, animals to be used in the study must be injected with a control virus encoding a nonpathogenic protein (i.e., GFP or other marker) to confirm that the observed phenotype is transgene-specific.

## 5. Cell Culture Models

Employing cellular models derived from human tissues mitigates the challenges posed by inter-species variations inherent in animal models. The development of representative experimental models based on adult cells is significantly hampered by the insufficient supply of high-quality postmortem tissue. Progress in stem cell research has addressed this limitation. In 2006, a group of Japanese scientists led by S. Yamanaka developed a method of reprogramming somatic cells into a pluripotent state by expressing four transcription factors: Oct3/4, Sox2, c-Myc, and Klf4 [249,250]. The resulting induced pluripotent stem cells (iPSCs) can be further differentiated into the desired cell types, including neurons and glia cells.

Initial neurological disease models using the iPSC technology focused on monogenic disorders or complex diseases with known causative mutations. These disorders include Parkinson’s disease [251,252,253], amyotrophic lateral sclerosis [254,255,256], smooth muscle atrophy [257,258], familial dysautonomia [259], Rett syndrome [260], schizophrenia [261], and others.

Human iPSCs have recently been derived from patients exhibiting both familial and sporadic Alzheimer’s disease. Skin and blood cells that are readily available are suitable for iPSC generation [262]. For example, nerve cells were derived from normal skin fibroblasts from an 82-year-old patient with a sporadic form of the disease [263]. The literature contains numerous reports detailing the generation of iPSCs from fibroblasts obtained from individuals with familial Alzheimer’s disease, attributed to a range of mutations [264,265,266]. The genomic diversity of these cells, sourced from Alzheimer’s patients, enables investigation into disease mechanisms and personalized drug screening [267,268,269] (Figure 4). Additionally, iPSC technology enables research unburdened by the gene overexpression artifacts characteristic of transgenic animal models.

Selected features of amyloid- and tau-related disease progression are reproduced in two-dimensional cellular cultures generated from induced pluripotent stem cells. Reprogrammed neuronal cultures derived from familial and sporadic Alzheimer’s disease patients exhibit elevated Aβ40/Aβ42 production and hyperphosphorylated tau compared with isogenic controls [265,270,271]. Several iPSC lines demonstrate the presence of further pathologies characteristic of Alzheimer’s disease. For example, the analysis of iPSC-derived neurons with the APPE693delta mutation revealed the intracellular accumulation of Aβ oligomers and indicators of cellular stress [270]. Subsequent research analyzed APP duplication (APPDp), a condition also correlated with early-onset, familial Alzheimer’s disease. Neurons exhibiting APPDp overexpression were determined to demonstrate augmented Aβ40 production and secretion, coupled with heightened GSK3β activity and increased phosphorylation of tau protein at threonine 231, a recognized pathological site [265]. Subsequent research exploring the London APP _V717I_ mutation’s effects on iPSC-derived neuronal cultures demonstrated the augmented production of Aβ42 and Aβ38, without a corresponding increase in Aβ40. Additionally, heightened β-secretase-mediated APP cleavage and elevated levels of total and phosphorylated tau protein were observed [271].

The analysis of presenilin 1 and presenilin 2 gene mutations using cell cultures has been the subject of numerous studies. Fibroblast-derived iPSCs from individuals with PS1 _A246E_ and PS1 _M146L_ mutations were procured [264,272]. Analyses revealed an association between the PS1 _L166P_ mutation and both a higher Aβ42–Aβ40 ratio and altered reactivity to γ-secretase modulators [273]. Mutations affecting other presenilin genes, including PS2 N141I, demonstrate a consistent outcome: increased Aβ42–Aβ40 ratios in iPSC-derived neuronal cultures and augmented Aβ42 synthesis [264,274].

Employing iPSC technology allows for the generation of cultures not only of neuronal cells but also of other neural tissues, thereby enabling research on the impact of diverse mutations and conditions on cellular function. In accordance with the findings presented in [275], astrocytes isolated from subjects carrying the PS1 _ΔE9_ mutation displayed enhanced Aβ production, atypical cytokine secretion, and elevated oxidative stress levels. An investigation into the effects of APOE4, PS1 _ΔE9_, and APP _Swe_ mutations revealed that APOE4 exerted the most significant impact on iPSC-derived microglia, diminishing phagocytosis and migration while augmenting cytokine release [276].

Two-dimensional cell cultures present limitations in modeling diverse cell interactions, complex tissue structures, vascular networks, and the blood–brain barrier [277,278]. A significant difference is observed in neuronal function and morphology between 2D and 3D models [279]. Moreover, the 2D models showed the absence of amyloid plaque and neurofibrillary tangle formation and neurodegeneration [280].

### 5.1. Three-Dimensional Modeling of Cell Cultures

In recent years, advances have been made in the development of more sophisticated Alzheimer’s disease modeling systems. Certain limitations of two-dimensional cell culture methodologies may be alleviated by the implementation of co- or triple-culture systems [281]. Three-dimensional cultures represent a more advanced approach. Model systems encompass those based on three-dimensional scaffolds, spheroids, or cerebral organoids cultured in suspension and those employing microfluidic technologies. Complex cultures allow some of the limitations of 2D approaches to be overcome. They reproduce the spatial and chemical complexity of living tissues in a more adequate way. Three-dimensional models offer the considerable advantage of utilizing extracellular matrix analogs, such as hydrogels (Matrigel, collagen, and fibrin), resulting in models that more closely mimic in vivo physiology [282]. Human iPSC-derived 3D models constitute a powerful methodology for pathology research and high-throughput drug screening, owing to their preservation of the complete human genetic background (including disease-associated mutations), capacity for in vitro expansion, and compatibility with diverse experimental manipulations and quantitative techniques [283]. In the following, we briefly review the various methods that are used for the 3D modeling of cell cultures. For more details, one is referred to the reviews [284,285].

#### 5.1.1. Three-Dimensional Framework Models

A straightforward method for transitioning from 2D to 3D cell cultures involves supplementing the culture medium with extracellular matrix components. The predominant scaffold for three-dimensional neuronal cell cultures is Matrigel. Nevertheless, research has explored alternative materials, as comprehensively reviewed in [286]. A 3D Matrigel-based culture of human neural stem cells transfected with APP _Swe_ × APP _Lon_ × PS1 _ΔE9_ was shown to produce extracellular amyloid beta plaque aggregation and intracellular tau aggregation in dystrophic neurites and cell bodies [287]. In the creation of framework-based models, careful consideration must be given to how the selected framework’s properties affect Aβ aggregation and cytotoxicity. Simpson et al. showed that all hydrogels under investigation affected the kinetics of Aβ aggregation [288]. For example, laminin, an important component of Matrigel, exhibited high affinity for Aβ and inhibited Aβ fibril formation [289]. This property must be taken into account during the evaluation of Matrigel-based Alzheimer’s disease models.

#### 5.1.2. Spheroids

Spheroids represent a prevalent modeling system for Alzheimer’s disease in contemporary research. Spheroid formation techniques are examined in [286]. The formation of spheroids from stem cell-derived lineages occurs naturally in a suitable culture medium and under appropriate conditions, irrespective of exogenous matrix addition. The engineering management of spheroids is less demanding than that of 3D frameworks [290].

The utilization of in vitro spheroid models has generated interesting findings pertinent to the investigation of familial and sporadic forms of Alzheimer’s disease [291]. In a cohort of five spheroids derived from sporadic cases, four exhibited a marked decrease in amyloid beta burden following BACE1 inhibitor treatment, with one spheroid demonstrating limited responsiveness to the therapy. Advanced analysis demonstrated that this cell line exhibited superior APP expression when compared with the other four cell lines [291]. These findings indicate potential interpatient variability in APP expression, influencing the dose-dependent efficacy of Aβ clearance therapies. The marked variability observed in patient-derived cultures renders prediction using animal models of Alzheimer’s disease infeasible. An analysis of hippocampal neurospheroids derived from iPSCs of patients harboring the London APP _V717I_ and PS1 _R278K_ mutations revealed elevated Aβ42 levels and Aβ42–Aβ40 ratios, synaptic protein deficits, and in the case of the APP mutation, substantial transcriptomic alterations and morphological abnormalities [292].

Another investigation yielded a spheroidal co-culture model utilizing neurons and astrocytes differentiated from the iPSCs of individuals with Alzheimer’s disease. Aβ aggregation within the resultant spheroids triggered caspase activation and subsequent cellular apoptosis. Conversely, neuroprotective agents demonstrated inhibitory effects. These experiments validate the use of 3D spheroids as a drug screening platform for Alzheimer’s disease [283].

Although spheroids have proven useful as Alzheimer’s disease models, inherent limitations persist. The deficiency of vascularization in spheroids is a significant drawback. Nutrient media and any drugs of interest cannot effectively penetrate the inner layers of the spheroid. Consequently, spheroid size and age are strictly limited, hindering their use in modeling age-related diseases [293].

#### 5.1.3. Organoids

The formation of cerebral organoids, in contrast to spheroids, often, but not always, uses a framework on which stem cells (embryonic, iPSCs) form structures similar in their structure to those of the corresponding organs [284,294]. This approach exploits the inherent self-organizing capacity of stem and progenitor cells, differing from spheroid generation techniques that pre-determine stem cell differentiation along a particular lineage before suspension culture to produce aggregates. The degree of external control over stem cell differentiation influences their self-organization, leading to structures that mimic different parts of the brain or the entire developing brain, including various neuron and astrocyte populations [295].

Several organoids modeling different types of Alzheimer’s have been created by using both patient iPSCs and genetic modification techniques. For example, amyloid aggregation followed by the hyperphosphorylation of tau protein was reproduced in organoids from the iPSCs of patients with familial Alzheimer’s disease with APP duplication [296]. Aggregates of Aβ and phosphorylated tau were observed in organoids in the presence of the PS1 _A246E_ mutation, with cell apoptosis being proportional to the accumulation of these aggregates [297]. Increased expression of the proinflammatory cytokines IL-6 and TNFα was also detected in organoids with the PS1 _M146V_ mutation [298]. Another study reported increased Aβ42–Aβ40 ratio, accumulation of amyloid plaques, Ser202/Thr205 hyperphosphorylation of tau protein, increased levels of cellular stress/apoptosis and senescence markers, and altered structural development in organoids with PS1 _A246E_ and PS2 _N141I_ mutations compared with organoids from iPSCs from healthy donors [299]. Organoids derived from patients with different mutations in PSEN1 and APP _V717I_ exhibited Aβ aggregates and increased levels of hyperphosphorylated tau, as well as significantly decreased levels of 5-hydroxymethylcytosine in DNA, compared with control organoids derived from cells from healthy donors [300]. The different forms of the APOE gene are central to understanding sporadic Alzheimer’s disease. Organoids containing the APOE4 allele were observed to exhibit increased apoptosis, more stress granules, and reduced synaptic integrity [301]. Furthermore, converting APOE4 into APOE3 isogenically before creating organoids lessened these effects.

The disadvantages of using organoids, as in the case of spheroids, are the necrosis of cells in deep layers due to insufficient diffusion of nutrients, the difficult diffusion of drugs inside the organoid, the absence of the vascular system involved in the clearance of neurotoxins, heterogeneity in batches of obtained organoids, the absence of the microglia and the blood–brain barrier involved in inflammation processes, the absence of connection with other organs, and the correspondence of the organoid to the early stages of brain development [293,302]. Protocols for initiating the development of vascular network and blood–brain barrier components and microglial cell differentiation in organoids in various ways have already been established. However, they still need to be refined and validated for the screening of drugs to treat Alzheimer’s disease [303].

#### 5.1.4. Microfluidic Systems: Organs-on-Chips

In 2003, microfluidic technology was proposed as the basis for methods to grow and study cultures of neural and other cells in microfluidic devices [304]. Microfluidic systems (chips) are plates with hollow channels containing living cells and tissues, all cultured within a flowing fluid. Certain devices can reproduce organ-level structures (Figure 5A). Figure 5B,C illustrate how the fluidic coupling of multiple organ chips generates multi-organ human body-on-chip systems that replicate whole-body physiology and drug delivery.

Employing a microfluidic platform addresses the shortcomings of alternative 3D cellular models, especially in the modeling of diverse cell-type interactions. A microfluidic chip facilitated the creation of a tri-culture model consisting of neurons (from a transduced neuronal progenitor cell line expressing mutant APP), astrocytes, and microglia. This model exhibited amyloid aggregation, elevated hyperphosphorylated tau, microglial activation, and proinflammatory cytokine/chemokine secretion [305]. A microfluidic system was utilized to construct an inflammatory model. This model included neurons with mutations in the APP and PSEN1 genes, along with astrocytes, microglia, and peripheral blood cells [306]. Compared with the controls, neuroglial cultures showed substantially elevated T-cell and monocyte infiltration, with CD8+ T cells demonstrating stronger activation of microglia, leading to exacerbated neuroinflammation and neurodegeneration.

Advancements in microfluidic technologies have significantly improved brain modeling capabilities. This progress is attributable to factors such as the asymmetric spatial organization of dendrites and axons [307], the capacity to model the blood–brain barrier (BBB), and the incorporation of interstitial flow dynamics. Investigating models incorporating the BBB is crucial to understanding the consequences of compromised integrity and evaluating its permeability to diverse pharmaceutical agents. The application of microfluidic technologies allows for the representation of the BBB under normal conditions [308,309], the computational modeling of neuroinflammation involving the BBB [310], and the analysis of compound transport across the BBB using human whole blood [311]. A microfluidic model of a BBB microvascular network was developed, integrating neurospheres derived from neural progenitor cells that possess Alzheimer’s disease-specific mutations within a compartmentalized system [312]. This model has revealed considerable changes in barrier permeability and morphology in the context of Alzheimer’s disease. Furthermore, amyloid β exhibited localized deposition within vascular networks following co-culture with Alzheimer’s disease-specific microtissues.

Three-dimensional (3D) cell culture systems, including spheroids, organoids, scaffold-based cultures, and microfluidic devices, provide distinct advantages as Alzheimer’s disease models. Although spheroids and organoids mimic the structural organization of in vivo brain tissue, they lack precise control over culture parameters, thus compromising reproducibility. The enhanced control over the cellular microenvironment offered by three-dimensional scaffolds often sacrifices physiological accuracy. Microfluidic platforms integrating the benefits of spheroid cultures and three-dimensional scaffolds may prove essential to advancing the development of next-generation Alzheimer’s disease models. These systems offer precise control over the composition and microenvironment, enabling the separation of distinct cell types within multi-chamber co-cultures and the regulation of physiological fluid flow, thereby enhancing the translational relevance of these human cell culture models of Alzheimer’s disease.

## 6. Computational Approaches to Modeling Alzheimer’s Disease: In Silico Methods

Active development is underway for databases, computational models, and simulations designed to represent human physiological and pathological processes. Computational models are applicable to both predictive modeling and hypothesis generation. These methods offer a means to replace or decrease reliance on in vivo studies, consequently reducing animal testing needs. Pharmacology extensively utilizes in silico methodologies for toxicology assessments and drug screening. This approach mitigates the temporal and financial burdens associated with pharmaceutical development [313]. Subsequently, we will examine computational methods for Alzheimer’s disease research.

### 6.1. Computational Modeling of Protein Interactions and Molecular Dynamics

In the last decade, computational modeling has proven to be a highly effective instrument in researching a wide array of human diseases, such as Alzheimer’s disease. Computational modeling offers valuable insights into molecular alterations and network dynamics underlying Alzheimer’s disease pathogenesis, potentially facilitating advancements in treatment [314]. The major patho-etiological characteristics of Alzheimer’s disease have been identified through the development of computational models using in silico techniques [315].

Computational models integrating tau and Aβ offer precise, quantitative insights into the aggregation processes of these proteins. These methods enabled the calculation of the free energy profile for the dissociation of a single tau monomer from the protofibril terminus in NFTs and the influence of Ser356 phosphorylation on this process [316]. This enabled a comprehensive analysis of the conformational changes during tau monomer integration into protofibrils.

The utility of molecular dynamic simulations extends to elucidating the mechanisms through which drugs interact with aggregated Aβ and tau proteins [317,318].

One must recognize that while useful, computational models are fundamentally grounded in mathematical algorithms. Such models offer novel perspectives, enhance comprehension, and integrate data from diverse sources; however, computational modeling remains insufficient as a substitute for empirical investigation [319,320].

### 6.2. Systematic Methodologies for the Analysis of Gene and Gene Expression Data

The past few decades have witnessed the accumulation of a vast quantity of genetic data from individuals diagnosed with Alzheimer’s disease. Bioinformatic approaches enable the analysis of extensive datasets derived from high-throughput experimental procedures. The integration of genetic and proteomic data may improve our comprehension of disease pathophysiology.

A significant challenge in researching complex diseases stems from their inherent heterogeneity. In particular, considerable variation in disease presentation is common among patients with complex illnesses. Furthermore, a complex network of interactions arises from the interplay of all genes, gene products, and small molecules. The variety of intermolecular interactions, such as those between proteins, proteins and DNA, and RNA, holds considerable importance. Alteration in a single gene may propagate through an interaction network, thereby influencing other network components. A similar Alzheimer’s disease phenotype resulting from varied causative agents implies that these agents converge upon a common point of cellular regulatory disruption [321]. Hence, the analysis of gene interactions is essential to the study of complex diseases.

Network construction methodologies are employed to investigate the interrelationships of biological macromolecules. Interaction networks encompass physical, functional, and regulatory interactions. Physical interaction networks offer insights into intermolecular interactions. The architecture of functional networks links genes exhibiting functional similarity or relatedness independently of any physical interaction. Direct and indirect regulatory relationships are represented within functional regulatory networks.

Numerous computational approaches for deriving functional interaction networks have been introduced [322]. Co-expression networks can be generated by calculating the correlation coefficients or mutual information between the gene expression profiles of each gene pair across different experimental conditions. More comprehensive functional networks can be constructed by integrating gene information with supplementary data sources, such as Gene Ontology annotations, genetic interaction data, and physical interaction data [323,324]. Cytoscape (San Diego, California, USA) provides a suitable platform for the visualization and annotation of networks.

A differential analysis of activated pathways in Alzheimer’s pathogenesis was conducted across six brain regions by using protein interaction weighting and boundary ranking estimation. This analysis revealed the presence of β-secretase, γ-secretase, APP, MAPT, APOE, LRP1, SNCA, CASP3, and CASP7 within the active pathways of all six regions under study [325]. Subsequently, an algorithm was created to analyze gene expression profiles and the protein interactions of the resultant gene products. The protein pathway network was assembled by integrating data from the Kyoto Encyclopedia of Genes and Genomes Pathways (KEGG) [326]. The synthesis of gene expression data pertaining to Alzheimer’s disease and protein interactome data resulted in the construction of a network featuring hub genes, the irregularities of which are associated with Alzheimer’s disease [327,328]. The analysis of this network revealed the involvement of mitogen-activated protein kinase (MAPK/ERK) and clathrin-mediated receptor endocytosis in Alzheimer’s disease development [329].

A subsequent study yielded a computational method for generating a ranked list of proteins significantly associated with Alzheimer’s disease. A comprehensive Alzheimer’s disease protein interaction subnetwork was produced through the application of this methodology. Analysis indicated that APP, PSEN1, and LRP1 were the three most prominent known associated proteins, while CTNNB1 (β-catenin), ranked 18th, constituted a novel finding [330].

To conclude, integrative systems biology approaches are demonstrably effective in improving the understanding of disease pathophysiology and identifying specific disease markers, including those relevant to Alzheimer’s disease. The application of network biology approaches has resulted in the identification of numerous novel genes and pathways relevant to Alzheimer’s disease, underscoring the efficacy of bioinformatics in the analysis of multivariate data.

### 6.3. Omics Technologies

Tissue sequencing provides an averaged representation of gene expression across all cells within a sample. This methodology proves beneficial to the assessment of differential gene expression within diverse brain regions [331]. Studies of Alzheimer’s disease and other neurodegenerative disorders using mass transcriptome analysis have revealed regionally distinct gene expression profiles [332,333]. Single-cell sequencing (SCS) technology affords a highly accurate description of cellular developmental trajectories and individual cell types [334].

Single-cell transcriptomic analysis yields transcriptional profiles for thousands to millions of individual cells, each exhibiting the expression of tens of thousands of genes. The fine-grained nature of these data enables investigation into specific cellular responses to pathogenesis. A considerable number of these data, however, demand robust computational methodologies for their interpretation [335,336].

Gene network dynamic modeling facilitates the extraction of biological information from single-cell transcriptomic data. Through gene network reconstruction, one can identify transcriptional regulators and their corresponding targets. Developing algorithms for single-cell transcriptome analysis to identify novel regulatory mechanisms presents a significant challenge. Genetic networks are extended and validated through the utilization of databases [337].

Single-nucleus RNA sequencing (snRNA-seq) has proven instrumental in elucidating aberrant gene expression profiles in Alzheimer’s disease research, particularly across diverse brain cell populations [338]. A pioneering large-scale study utilizing snRNA-seq on postmortem brain tissue from Alzheimer’s disease patients (n = 48) revealed myelination deficits across multiple cell types within the prefrontal cortex, correlating with disease severity. Large transcriptional changes were shown to occur early in the disease before the development of severe pathologic features. The authors also presented evidence of APOE, an Alzheimer’s disease risk factor, being upregulated in microglia and downregulated in astrocytes. This finding highlighted the superiority of scRNA-seq methodologies compared with bulk RNA sequencing [338].

While high-quality single-cell transcriptome data acquisition is now a standardized procedure, its subsequent interpretation and validation remain significant and complex challenges. A crucial step in data validation and interpretation involves developing cell-based mathematical models that integrate data on genetic regulator expression within cells and the governing principles of intercellular communication, cell mechanics, and transport processes. Current technological limitations prevent the direct application of complete transcriptome datasets to the mathematical modeling of morphogenesis due to the extensive quantity of data points. Consequently, the identification and incorporation of key genetic and metabolic distinctions between cell types within models is crucial. Although single-cell studies offer a more detailed understanding of the cellular responses in Alzheimer’s disease, the complexity of the disease requires the consideration of multiple datasets to fully capture its heterogeneity.

### 6.4. Artificial Intelligence Methods in Alzheimer’s Disease Research

Artificial intelligence (AI) is a valuable tool for analyzing large datasets. First of all, AI in medicine is used to improve the quality of disease diagnostics, as well as predicting the course of the disease in a particular patient and selecting an appropriate treatment method [339]. Such tasks are aimed at identifying patterns in datasets such as neuroimaging data; genomic, proteomic, and metabolomic data; and cognitive test results. For this purpose, various methods of machine learning (ML) are used, including artificial neural networks with different architectures (e.g., feed-forward neural networks, convolutional neural networks, recurrent neural networks, autoencoders, graph convolutional networks, and others) [340,341,342].

By using traditional ML methods such as support vector machines and random forests, neuroimaging data can be used to distinguish Alzheimer’s patients from others [343,344,345]. For example, a new MRI biomarker for the onset of Alzheimer’s disease was developed by using semi-supervised learning, which significantly improved diagnostic accuracy [346]. However, traditional ML methods often rely on manual feature extraction, which is a labor-intensive process [347]. Deep learning (DL) is suitable for the analysis of high-dimensional datasets. DL technologies have allowed for the development of various models to diagnose and predict the development of Alzheimer’s disease by using both neuroimaging data (MRI and PET) and other patient data [342,348,349,350,351]. Various databases are used to train the models and test their effectiveness, including data from the Alzheimer’s Disease Neuroimaging Initiative (ADNI), the National Alzheimer’s Coordinating Center (NACC), the Australian Imaging Biomarkers and Lifestyle Study of Ageing (AIBL), the Open Access Series of Imaging Studies (OASIS), and others. DL and in particular deep generative models are also used to generate drug molecules de novo, for example, by using generative adversarial networks (GANs), and evaluate the properties of different molecules [352].

Explainable Artificial Intelligence (XAI) aims to improve the interpretability of AI systems by making decision-making processes more transparent to the user [353,354]. XAI methods help to identify the most important features and biomarkers in the diagnosis of Alzheimer’s disease, such as white matter hyperintensities [355] and brain atrophy patterns [356].

The Hopfield network is a type of recurrent artificial neural network (ANN) [357]. Its use was an early attempt to model cognitive processes such as memory and pattern recognition, and it laid the foundation for future developments in ANNs, influencing modern networks. The Hopfield network has been used to model stages of Alzheimer’s disease with memory loss [358].

Currently, there are many projects to study Alzheimer’s disease by using AI methods. For example, an IBM research study is one of the first large studies to use AI to predict the possible onset of Alzheimer’s disease in healthy individuals based on linguistic data [359]. AI-Mind is a European initiative to develop AI tools for early diagnosis based on MRI and cognitive tests [360].

Thus, AI has great potential in the field of Alzheimer’s disease with a wide range of applications. As technology advances and research progresses, AI is expected to continue to make significant breakthroughs and contribute to advances in human health.

## 7. Conclusions

Animals continue to be widely employed in the investigation of Alzheimer’s disease. Animal models offer a distinct advantage in providing an in vivo system for evaluating the general toxicity of new therapeutic agents and conducting cognitive studies. Although animal models have been continuously improved over the past decades, there is still no animal model that perfectly mimics both the pathological features and the cognitive impairment observed in patients with Alzheimer’s disease. Unfortunately, animal models do not reflect important pathological features of the human disease and do not adequately model the complex genetics associated with sporadic Alzheimer’s disease [361]. Also, the utilization of animals in biomedical research presents certain ethical challenges. Collaborative initiatives are in progress to cultivate superior, more insightful, and predictive models.

Recent developments indicate a significant movement towards anthropocentric approaches utilizing cellular and mathematical models. In contemporary research, human iPSC-derived central nervous system cells serve as indispensable tools for investigating disease mechanisms. Human neural culture offers significant potential for predicting the contribution of individual genetic and cellular phenotypic variation to the pharmacologic response at clinically relevant levels.

Developing sophisticated multicellular models is crucial to Alzheimer’s disease research. A highly promising method for generating multicellular models is offered by microfluidic technology. Through the application of these technologies, the complex and multi-structured brain can be fractionated into its constituent elements, thereby facilitating research into specific intercellular interactions and the tissue microenvironment. Furthermore, this allows for the integration of distinct brain subsystems, such as the blood–brain barrier (BBB), enabling the study of transport mechanisms across the BBB and the targeted delivery of drugs to the brain. The application of microfluidic technologies holds significant potential for preclinical in vitro studies, thereby offering an alternative to animal testing in drug and nanocarrier screening, as well as personalized medicine research.

A comprehensive understanding of disease pathogenesis may be facilitated by employing “omics”-based approaches, enabling genome-wide or proteome-wide screening to detect disrupted networks. In this regard, the incorporation of in silico research offers significant potential. Computational modeling may contribute to the advancement of personalized medicine for Alzheimer’s disease. Disease progression prediction and the design of tailored treatment plans are enabled through the integration of individual patient data, genetic profiles, and environmental factors within computational models. Computational analysis provides a means to examine the diverse nature of Alzheimer’s disease, given the observed differences in patient clinical features and disease progression. It is imperative to remember that integrating various modeling approaches is crucial to a complete understanding of pathogenic mechanisms.

Over the past few decades, dementia research has provided a deeper understanding of how Alzheimer’s disease affects the brain. Today, researchers continue to search for more effective treatments, as well as measures that could prevent Alzheimer’s disease and improve the health of patients. Many projects have been launched to investigate the mechanisms underlying the disease and help patients with neurodegenerative diseases, including Alzheimer’s. Existing projects are reviewed in detail in [362].

## Figures and Tables

**Figure 1 brainsci-15-00486-f001:**
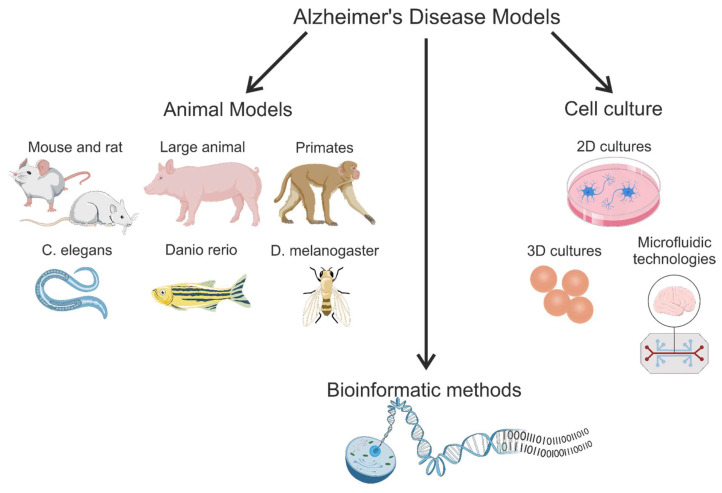
The major models relevant to Alzheimer’s disease considered in this review.

**Figure 2 brainsci-15-00486-f002:**
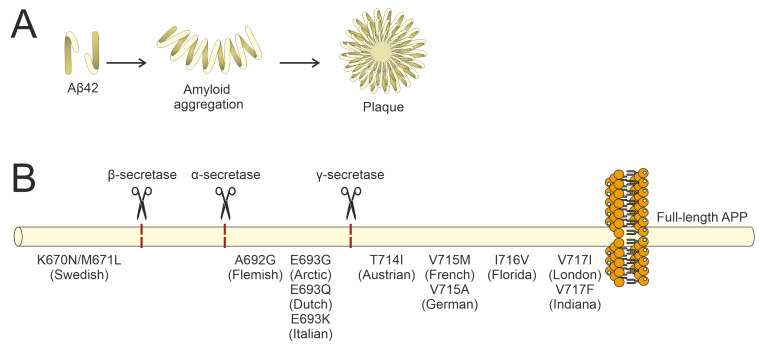
(**A**). The self-assembly of the less soluble peptides (Aβ42) into fibrils, resulting in the formation of senile plaques. (**B**). Generation of Aβ fragments from APP. Mutation sites on the longest APP isoform are specified.

**Figure 3 brainsci-15-00486-f003:**
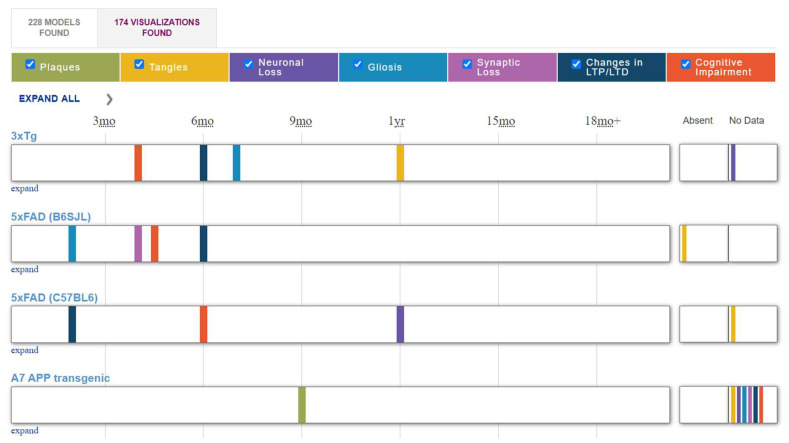
The illustration of a comparative analysis of various genetically modified mouse models pertinent to Alzheimer’s disease, drawing upon the resource located at [73].

**Figure 4 brainsci-15-00486-f004:**
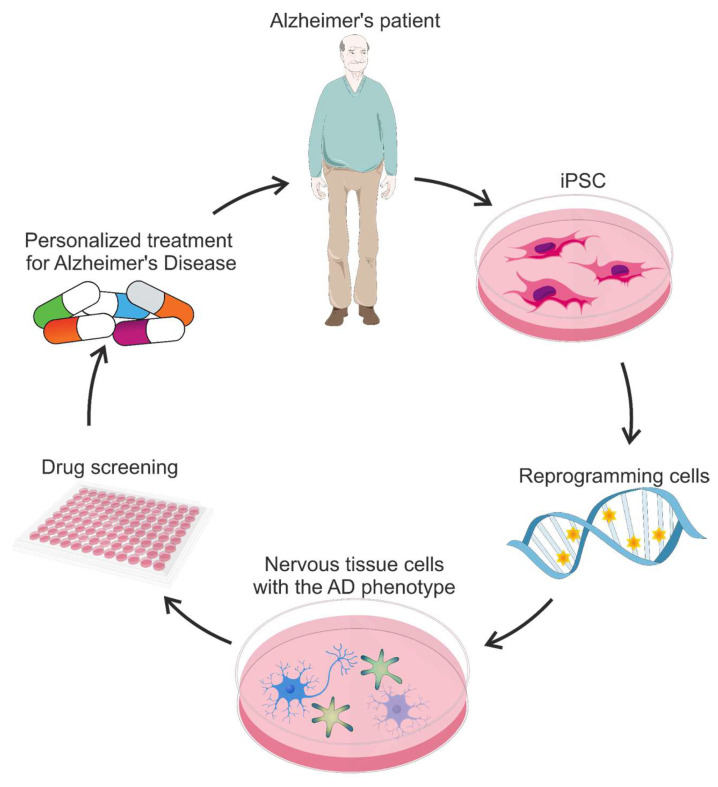
A general scheme of studies using induced pluripotent stem cells (iPSCs) derived from a patient to model the pathology of Alzheimer’s disease. The patient’s somatic cells (blood, skin, etc.) are reprogrammed in vitro into iPSC colonies. Techniques for the differentiation of iPSCs produce nerve cells that include neurons, microglia, astrocytes, and oligodendrocytes, as well as neural progenitor cells. Patient iPSCs are used to derive neural tissue cells with an Alzheimer’s disease phenotype to be used for screening new therapeutics.

**Figure 5 brainsci-15-00486-f005:**
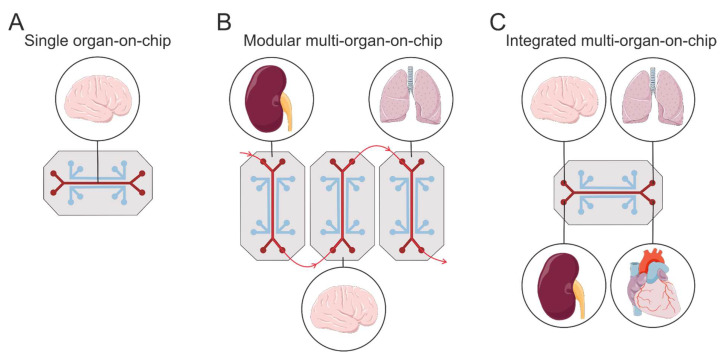
Diverse microfluidic modeling methodologies. (**A**) A model of a single organ, such as the brain. (**B**) A model based on the interconnection of several modules to study the interactions between different organs. (**C**) An integrated platform combining multiple organ systems on a single chip.

## Data Availability

No new data were created or analyzed in this study.

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
