# Peer review of "Modeling Alzheimer’s Disease: A Review of Gene-Modified and Induced Animal Models, Complex Cell Culture Models, and Computational Modeling"

_brainsci, 2025, doi:10.3390/brainsci15050486_

Round 1

Reviewer 1 Report

Comments and Suggestions for Authors

I hope the following comments allow you to consider the reader's facilitation in their readings and improvements in guiding their thoughts:
There is a little more information in Figure 3; it would be helpful to give a general and quantitative overview of the figure for better and smoother understandings.

In Figure 2, please add a timeline-based discovery of the generation of Aβ fragments from APP, which allows readers to have an overview of its historical aspects. Call it Section C in the same picture.

Please place the visualized design of the location regarding the injections in the following section: 4.2. Stereotactic administration of various forms of tau and Aβ.

I recommend giving an overview of the products that have been tested and have been named as successful medications in the case of clinical trial records.

I also recommend saying how new technologies like artificial intelligence could help these models. In which aspect?

Also give a little section that covers startup companies that are working in this area and how they helped in this topic, which is almost in alignment with your current dissections. In other words, how are private sections advancing this area of science?

Open a little section, and I think it is better to call it the dilemma of behavior or cognition. It might be good to cover this idea that discrepancy in the data could be related to the animal behavior tests simplicity and lack of good measures in animal behavioral science, which is not able to shift this topic ahead. It is nice to guide readers toward new behaviors that could help researchers to have better interaction about the impact of, e.g., injected APPs. This topic is really controversial, and I think it is better to have a little section.

Author Response

Dear Reviewer!

We have tried to eliminate all the shortcomings that you pointed out in your comments.

Comment 1: I hope the following comments allow you to consider the reader's facilitation in their readings and improvements in guiding their thoughts:
There is a little more information in Figure 3; it would be helpful to give a general and quantitative overview of the figure for better and smoother understandings. 

Response 1:  We have added a clarification to the manuscript text for Figure 3. See lines 198-200.

Comment 2: In Figure 2, please add a timeline-based discovery of the generation of Aβ fragments from APP, which allows readers to have an overview of its historical aspects. Call it Section C in the same picture.

Response 2: Thank you for your valuable comment and suggestion to add a timeline to Figure 2 showing the history of the discovery of the generation of Aβ fragments from APP. Two fragments play a major role in the pathogenesis of Alzheimer's disease: Aβ40 and Aβ42. The Aβ40 peptide was isolated from the cerebral vessels of Alzheimer's patients and described in 1984. In 1992 it was shown that Aβ42 is the dominant form in the brains of patients with the familial form of the disease. In the 1990s, with the development of mass spectrometry and sequencing, the structures of these peptides were elucidated and it was shown that a shift in processing towards Aβ42 increased aggregation and neurotoxicity.

To address your comment, we have added this information to the text of the manuscript. See lines 140-142.

Comment 3: Please place the visualized design of the location regarding the injections in the following section: 4.2. Stereotactic administration of various forms of tau and Aβ. 

Response 3:  We have added this. See lines 5503-555

Comment 4: I recommend giving an overview of the products that have been tested and have been named as successful medications in the case of clinical trial records.

Response 4:  Alzheimer's disease is one of the most difficult targets for pharmacotherapy. Over the past 20 years, more than 98% of clinical trials for Alzheimer's disease drugs have failed. Currently, a combination of approaches is usually used to develop therapeutic drugs. A review of drugs for Alzheimer's disease is unfortunately too complex to be covered in this review.

Comment 5: I also recommend saying how new technologies like artificial intelligence could help these models. In which aspect? 

Response 5: We have added a new section 5.4, which is dedicated to artificial neural networks and artificial intelligence.

Comment 6: Also give a little section that covers startup companies that are working in this area and how they helped in this topic, which is almost in alignment with your current dissections. In other words, how are private sections advancing this area of science? 

Response 6:  We have added information about Alzheimer's research projects (see lines 1077-1079). We also mentioned projects using AI (see lines 1026-1030).

Comment 7: Open a little section, and I think it is better to call it the dilemma of behavior or cognition. It might be good to cover this idea that discrepancy in the data could be related to the animal behavior tests simplicity and lack of good measures in animal behavioral science, which is not able to shift this topic ahead. It is nice to guide readers toward new behaviors that could help researchers to have better interaction about the impact of, e.g., injected APPs. This topic is really controversial, and I think it is better to have a little section. 

Response 7: We have listed limitations of the studies related to differences in pathological features and cognitive impairment in animal models and in patients with Alzheimer's disease. See lines 1037-1043.

Thank you on the behalf of co-authors of the manuscript!

Reviewer 2 Report

Comments and Suggestions for Authors

General Comment

This review article presents a broad and updated analysis of the main models used in the study of Alzheimer's disease, including genetically modified animal models, induced models, complex cell cultures, and computational modeling. It is a relevant and important work for the scientific community, as it provides an integrated overview of available methods for investigating the etiological and pathophysiological mechanisms of the disease, while also highlighting emerging approaches such as microfluidic technologies and bioinformatics. The review successfully updates researchers on the state of the art in Alzheimer's disease modeling and suggests promising directions for future research, especially in the context of personalized models and advanced computational techniques. However, before acceptance, I consider it necessary to revise several key points, detailed below.

Specific Comments

Comment 1: As this is a review article, it is recommended to clearly indicate this in the title.

Suggested revised title:

"Modeling Alzheimer's Disease: A Review of Gene-Modified and Induced Animals, Complex Cell Culture Models, and Computational Modeling"

Comment 2: It is essential to include a specific section describing the methodology for obtaining bibliographic references. The authors should clarify:

- Whether a recognized method such as PRISMA or PICO was used.

- The type of review: systematic, scoping, narrative, etc.

- Inclusion/exclusion criteria, databases used, language, publication period, and other transparency elements.

Comment 3: The introduction needs to clearly state:

- Which previous similar reviews exist, and up to what year they covered.

- What scientific gaps this new review aims to fill.

- The main findings and contributions of the review.

- How this work advances the current knowledge.

Comment 4: At the end of the introduction, include a transitional paragraph providing an overview of the sections covered in the article. This helps guide the reader and improve the flow.

Comment 5: References to websites (e.g., lines 165 and 166) must be included in the reference list and cited in the text according to the journal's formatting style.

Comment 6: To enrich the paper on etiology and pathogenesis, I suggest including the following studies:

- eLife. https://doi.org/10.7554/eLife.25659
- Cell Reports. https://doi.org/10.1016/j.celrep.2019.08.103

Comment 7: To strengthen the section on computational modeling, consider including the following papers:

- Neurocomputing. https://doi.org/10.1016/j.neucom.2025.129967

Comment 8: In vivo, in vitro... Must be in italic.

Author Response

Dear Reviewer!

Thank you for your valuable feedback. We have tried to eliminate all the shortcomings that you pointed out in your comments.

Comment 1: As this is a review article, it is recommended to clearly indicate this in the title.

Suggested revised title: "Modeling Alzheimer's Disease: A Review of Gene-Modified and Induced Animals, Complex Cell Culture Models, and Computational Modeling"

Response 1: Thank you for your suggestion. We have changed the title.

Comment 2: It is essential to include a specific section describing the methodology for obtaining bibliographic references. The authors should clarify:

- Whether a recognized method such as PRISMA or PICO was used.

- The type of review: systematic, scoping, narrative, etc.

- Inclusion/exclusion criteria, databases used, language, publication period, and other transparency elements.

Response 2: This manuscript is a narrative review that summarizes the main points and important information from various sources. Animal models are still widely used to study Alzheimer's disease. However, other approaches are needed to model this neurodegenerative disease. We wanted to highlight all potential avenues for Alzheimer's disease research. In our review, we did not use strict inclusion or exclusion criteria and included sources with relevant information available in the English-language PubMed database through 2025.

We have included this information in the introduction, see lines 103-105.

Comment 3: The introduction needs to clearly state:

- Which previous similar reviews exist, and up to what year they covered.

- What scientific gaps this new review aims to fill.

- The main findings and contributions of the review.

- How this work advances the current knowledge.

Response 3: Existing reviews usually focus on only one of the methods for modeling Alzheimer's disease. However, an in-depth study of the mechanisms of pathogenesis can only be achieved by combining different modeling methods. Integrating different modeling techniques facilitates in-depth analysis and understanding of pathogenic mechanisms. This review aims to systematize the knowledge about modeling methods. It also considers modern approaches such as the use of iPSC and the creation of complex cellular systems and the development of mathematical models. In contrast to animal models, these approaches are more human-centered, have great potential and can replace some animal experiments. We have added this information in the Introduction, see lines 106-114.

Comment 4: At the end of the introduction, include a transitional paragraph providing an overview of the sections covered in the article. This helps guide the reader and improve the flow. 

Response 4: We've added this. Look at the lines 115-129

Comment 5: References to websites (e.g., lines 165 and 166) must be included in the reference list and cited in the text according to the journal's formatting style.

Response 5: We have fixed this. See lines 198, 204 and Ref. 73.

Comment 6: To enrich the paper on etiology and pathogenesis, I suggest including the following studies:

- eLife. https://doi.org/10.7554/eLife.25659
- Cell Reports. https://doi.org/10.1016/j.celrep.2019.08.103

Response 6: We have mentioned the role of AICD in Alzheimer's disease. See lines 136-139, 144-146 and Ref. 26-27.

Comment 7: To strengthen the section on computational modeling, consider including the following papers: 

- Neurocomputing. https://doi.org/10.1016/j.neucom.2025.129967

Response 7: We have added information and a link to the article. See Ref 358

Comment 8: In vivo, in vitro... Must be in italic.

Response 8: We fixed this

Thank you on the behalf of co-authors of the manuscript!

Round 2

Reviewer 1 Report

Comments and Suggestions for Authors

Dear Authors,
I would like to see more tables and an additional picture related to the injection, which was not included in the provided version. Some of the required items have already been addressed, and I believe the editor should handle the remaining work. Your review has several strong aspects, and I prefer to focus on the positive ones. I hope this review contributes to advancing Alzheimer’s community.

Author Response

Thank you very much for your careful reading of our work and your valuable recommendations. We appreciate your desire to improve the article and your support for its strengths. In our opinion, we have included in the manuscript all the important points related to the modeling of Alzheimer's disease. We hope that the manuscript is complete. Adding tables and figures will increase the size of the manuscript, which may not be advisable.

Reviewer 2 Report

Comments and Suggestions for Authors

I appreciate the authors' attention and care in addressing the suggestions provided. I recommend the acceptance of the paper in its current form.

Author Response

We thank the reviewer for his valuable comments.